# Inter-Habitat Variability in Parrotfish Bioerosion Rates and Grazing Pressure on an Indian Ocean Reef Platform

**Robert T. Yarlett [1,\*], Chris T. Perry [1,\*] [ID], Rod W. Wilson [2] and Alastair R. Harborne [3]**

1    Geography, College of Life and Environmental Sciences, University of Exeter, Exeter EX4 4RJ, UK
2    Biosciences, College of Life and Environmental Sciences, University of Exeter, Exeter EX4 4QD, UK;
     R.W.Wilson@exeter.ac.uk
3    Institute of Environment and Department of Biological Sciences, Florida International University,
     North Miami, FL 33181, USA; aharborn@fiu.edu
\*    Correspondence: yarlett.r@gmail.com (R.T.Y.); C.Perry@exeter.ac.uk (C.T.P.)

**Abstract:** Parrotfish perform a variety of vital ecological functions on coral reefs, but we have little understanding of how these vary spatially as a result of inter-habitat variability in species assemblages. Here, we examine how two key ecological functions that result from parrotfish feeding, bioerosion and substrate grazing, vary between habitats over a reef scale in the central Maldives. Eight distinct habitats were delineated in early 2015, prior to the 2016 bleaching event, each supporting a unique parrotfish assemblage. Bioerosion rates varied from 0 to 0.84 ± 0.12 kg m$^{-2}$ yr$^{-1}$ but were highest in the coral rubble- and *Pocillopora* spp.-dominated habitat. Grazing pressure also varied markedly between habitats but followed a different inter-habitat pattern from that of bioerosion, with different contributing species. Total parrotfish grazing pressure ranged from 0 to ~264 ± 16% available substrate grazed yr$^{-1}$ in the branching *Acropora* spp.-dominated habitat. Despite the importance of these functions in influencing reef-scale physical structure and ecological health, the highest rates occurred over less than 30% of the platform area. The results presented here provide new insights into within-reef variability in parrotfish ecological functions and demonstrate the importance of considering how these interact to influence reef geo-ecology.

**Keywords:** coral reefs; parrotfish; bioerosion; grazing; ecological functions

## 1. Introduction

Coral reefs are built and shaped, both structurally and ecologically, by the organisms that inhabit them [1]. Carbonate production (e.g., by scleractinian corals and coralline algae) and bioerosion (e.g., by fish and urchins) are especially important controls on reef growth potential and topographic complexity, thereby influencing wave energy regimes and habitat provision for many commercially important species [2–5]. Along with bioerosion, grazing by fish and urchins is important, because it impedes build-up of foliose algal biomass and conditions the composition of turf algae assemblages [6,7]. In turn, this can increase juvenile coral survival rates [8], reduce partial coral mortality and disease [9,10], and increase reef resilience [11]. Grazing therefore has an indirect influence on reef carbonate production rates [12,13].

On coral reefs, parrotfish are key grazers, and some species, which have musculoskeletal jaw architectures that are particularly well adapted for biting into reef substrates, are also important substrate bioeroders [14–16]. There is now strong evidence that parrotfish target protein-rich cyanobacteria living on and within the reef framework as their primary food source, at least in the Indo-Pacific [17,18]. In the process of feeding, primarily on dead coral and rubble substrates, parrotfish also remove and

consume algal turfs [19–21]. Parrotfish also erode and ingest carbonate substrate to access endolithic food resources, although some species do this more than others [18]. The ingested substrate is broken down by modified gill arch elements, collectively known as the pharyngeal mill [22,23]. This material is processed in the gut along with organic matter and egested into the environment as sediment [20–25]. In some regions, parrotfish bioerosion and the resultant sand egestion has been reported to dominate reef sediment production [26–28]. Bioerosion and grazing thus play important roles in overall reef carbonate production and cycling processes, and act as a "top-down" influence on reef ecological and physical structure [16–30].

However, grazing pressure (defined here as the total surface area of parrotfish bites, expressed as a proportion of grazable substrate area per year—as an indicator of the area of substrate bitten per year) and bioerosion rates (the mass of reef substrate eroded per year) can vary significantly among species, fish size classes, and between "scraper" (where bites are restricted to the removal of substrate surface material) and "excavator" (where bites remove chunks of substrate material) species [22–32]. These issues become important for understanding habitat-scale parrotfish ecological functions, because parrotfish assemblages (as with other taxonomic groups) can vary markedly between habitats or along gradients of structural complexity [33–35]. For example, comparisons between the Red Sea, Arabian Sea, and Arabian Gulf revealed marked regional differences in parrotfish bioerosion rates and grazing pressure because of variations in species assemblage [36].

Whereas many studies have investigated the top-down influence of parrotfish on reef habitats, our knowledge of how parrotfish assemblages in different reef habitats vary in their ecological functions is limited. Hoey and Bellwood (2008) [25] demonstrated how the ecological functions of whole parrotfish assemblages vary between inner, mid, and outer reef environments on the Great Barrier Reef. However, many reef systems can have a very different habitat structure compared to barrier reefs, including atoll reef platforms (an isolated reef within a larger atoll structure), fringing reefs, or systems with large lagoons, reef flats, seagrass meadows, or mangrove forests. These systems differ in terms of the spatial extent of their main habitats, the species that they support, their benthic community composition, geomorphology, and the extent to which they are influenced by external factors such as terrestrial nutrient and sediment inputs. These systems have received little attention, particularly in the central Indian Ocean.

In many regions, it is difficult to tease apart the influences of fishing pressure and habitat type on parrotfish assemblages [37]. However, the Maldives represents an example environment where parrotfish are not a main fishery target [38], making it a useful natural laboratory for examining the natural influence of habitat type on parrotfish ecological functions. Here, we examine the contributions of parrotfish species to grazing pressure and bioerosion rates across an atoll-edge reef platform in the central Maldives (Vavvaru Island, Lhaviyani Atoll). Specifically, we address the following questions: (1) How does total grazing pressure and bioerosion rate vary among habitat types as a function of parrotfish assemblage? (2) What are the dominant species and size contributors to these geo-ecological functions? Empirical data on these issues are needed to understand how these ecological functions are likely to respond to ongoing environmental change [25,28,39].

## 2. Materials and Methods

### 2.1. Study Site

Field data were collected in February 2015 from an atoll-edge reef platform, Vavvaru, Lhaviyani Atoll, in north-central Maldives (5°25′5.0″ N, 073°21′14.0″ E; Figure 1). The reef platform at Vavvaru comprised eight distinct marine habitats, making the site ideal for examining variation in parrotfish ecological functions among reef habitats. Habitats were delineated in-situ based on field observations and measures of the rugosity, substrate characteristics, and benthic communities, and the spatial extent of each habitat was then estimated from satellite imagery and ground truthing (Figure 1, as described in Perry et al., 2017 [28]).

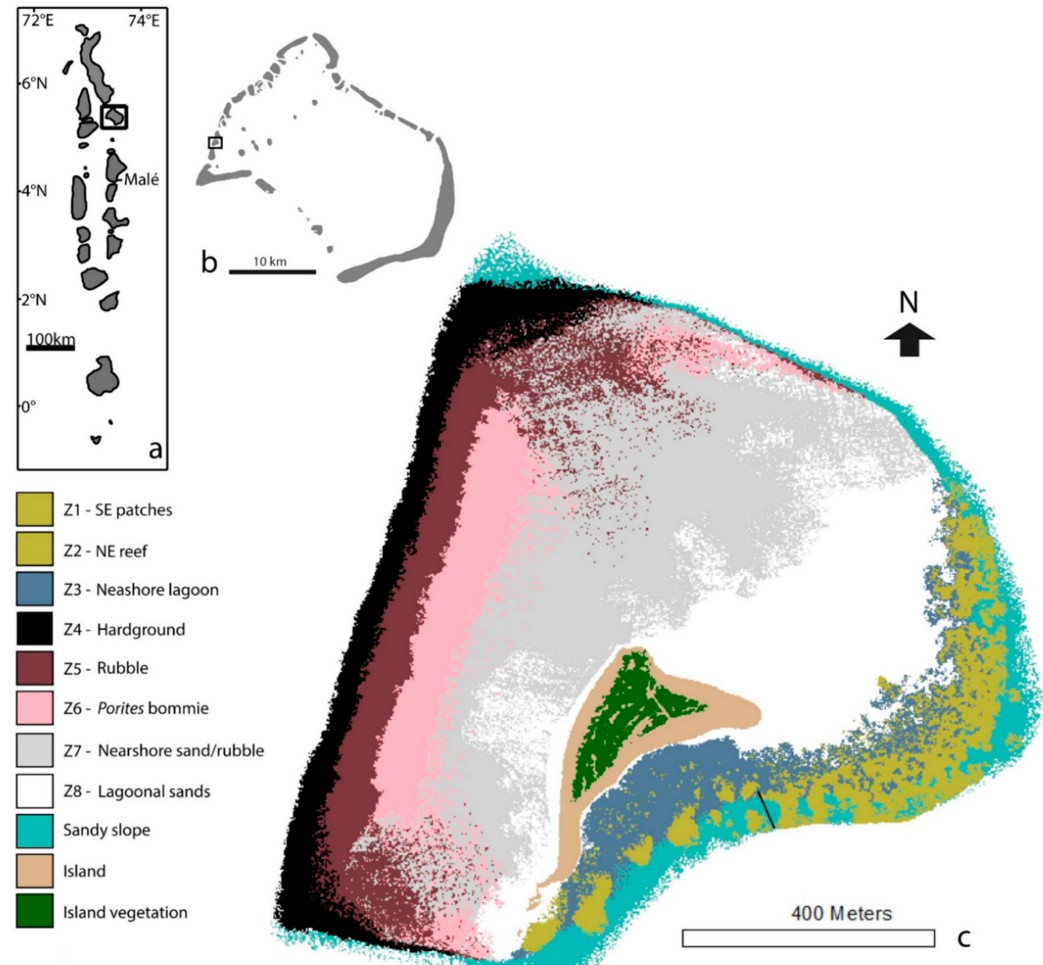

**Figure 1.** (**a**) Position of Lhaviyani Atoll in the Maldives. (**b**) Position of Vavvaru on Lhaviyani Atoll. (**c**) Habitat map of Vavvaru produced from Quickbird imagery of western Lhaviyani Atoll taken on 9 July 2008 (provided by DigitalGlobe Foundation; http://www.digitalglobefoundation.org/) and ground validated points. See Perry et al. (2017) [28] for original publication and position of ground points. Note that the southeast (SE) patches (Z1) and northeast (NE) reef (Z2) could not be differentiated during analysis of satellite data and are presented as the same color here. However, the black line marks the division of NE reef and the SE patches, which were treated as separate habitats (based on in-situ observations and geo-ecological data collection) in the estimates of bioerosion rate and grazing pressure in the present study.

The far western edge of the Vavvaru reef platform consists of a hardground (limestone pavement) habitat (Z4) with reasonably high coral cover (18.81%) at 4–6 m depth, characterized by a very steep wall marking the margin of Lhaviyani atoll. Moving east, the hardgrounds transition into a gently sloping coral rubble and *Pocillopora* spp.-dominated habitat (from ~5 m at the hardground/rubble transition, sloping gently up to a shallow rubble ridge at ~1-m depth, Z5) and then into a shallow (<2 m) limestone pavement habitat dominated by *Porites* spp. bommies (Z6). The central area of the platform is made up of the two largest and relatively featureless marine habitats: a sand and coral rubble habitat (Z7), and an extensive sandy lagoon (Z8) situated to the north of Vavvaru Island. The island itself is situated off-center, toward the southeast of the platform. To the northeast of the platform is an *Acropora* spp.-dominated reef habitat (Z2), which becomes more fragmented toward the south and transitions into patch reefs (Z1) separated by irregular sand channels. Both of these reef habitats are shallow on their nearshore sides (<2 m) but form part of the reef slope at the eastern edge of the platform, where coral cover extends down to ~8 m (and deeper in some parts of Z2). Between Vavvaru

Island and these eastern reef habitats is a lagoon (Z3), comprised predominantly of sand, but with small (<10 m$^2$) scattered patch reefs, which increase in frequency toward the eastern reef habitats. A sand talus on the eastern slope extends into the atoll lagoon. Images of these habitats can be found in the Electronic Supplementary Material (Figure S1).

*2.2. Remote Underwater Video (RUV) Surveys*

Parrotfish were surveyed using Remote Underwater Video (RUV) to estimate the extent of parrotfish ecological functions in a given area of reef per time period (similar to a recent approach taken by Streit et al., 2019 [40]) rather than estimating grazing pressure and bioerosion rates from visual census data. Remote Underwater Video is considered to be a useful approach to quantify parrotfish ecological functions, because the method is designed to observe fish activity over an area of reef over a given period of time rather than estimate them from parrotfish density data, where there is a risk of over- or under-counting [41]. In addition, the long survey time (totaling over 200 h) and lack of human presence allows the contributions of rare and shy, but possibly ecologically important, species to be detected, thereby avoiding some of the problems that can arise with conventional visual survey techniques [42].

A range of Go-Pro Hero 4, 3+, and Intova Sport HDII cameras were mounted onto polyvinyl chloride (PVC) frames and deployed for a minimum of 1 h to capture both common and rare species [43]. In each habitat, 15 RUVs were randomly deployed, ensuring they were at least 20-m apart. Deployments spanned the entire length of the parrotfish feeding day (~06:30–18:30) with six replicates in the morning (sunrise–11:30), three at midday (11:30–14:00), and six in the afternoon (14:00–sunset). The camera was randomly redeployed in a new location in each time bin, so all 15 replicates were deployed in new locations. Four 50 cm scale bars with 5 cm increments were placed at 1 m intervals up to 4 m in front of the camera. These were removed ~30 s into each recording to avoid unnecessary disturbance to fish behavior. A screen overlay was then used to mark the position of the scales during data collection and allow estimates of fish size.

Analysis of each video began once the scale bars were removed and after allowing 2 min for observers to leave the area. All parrotfish entering the field of view within 4 m (the furthest scale bar) of the camera were recorded at their entry time, identified to species level and life phase, and assigned to one of the following size classes: <15 cm, 16 to 30 cm, 31 to 45 cm, >46 cm. Each species was also described by its primary functional group (excavator, scraper, browser) as defined in Bellwood and Choat (1990) [22]. A pilot study conducted prior to fieldwork revealed that classifying objects (PVC pipes) into 15-cm size categories using this method was correct 98% of the time, regardless of angle to the camera. The 4-m distance from the camera limit was chosen to ensure that visibility and distance of fish from the camera did not interfere with species identification. Juvenile parrotfish (which were rarely larger than 15 cm) were recorded without species information because of challenges associated with accurate visual identification. A dataset of ~3500 recordings of parrotfish video entry and exit times was used to estimate a mean parrotfish residence time in the survey area.

The videos were used to record the total number of parrotfish sightings of all species and size classes in a defined area of reef, over a given period of time, regardless of whether it was the same or different individuals entering the area. Using the total duration that each size class of each species spent in the area, and the known bioerosion rate and grazing pressure for those parrotfish, total parrotfish bioerosion rates and grazing pressure for the survey area and duration were estimated.

*2.3. Inter-Habitat Variability in Total Parrotfish Bioerosion Rates*

To estimate overall parrotfish bioerosion rates in each habitat, local species- and size class-specific bite rates and grazing scar metrics were extracted from Yarlett et al. (2018) [44] for six of the most abundant and representative species at Vavvaru: *Chlorurus sordidus, Chlorurus strongylocephalus, Scarus frenatus, S. rubrioviolaceus, S. niger*, and *S. psittacus*. Rates for other species were assumed to match the most closely related species (based on Choat et al. 2012 [45]), or species with the most comparable

morphology (see Table 1). Parrotfish bite rates were assumed to remain relatively consistent throughout the year because there is little seasonal variation (<30 min) in daylight hours or water temperature (<2 °C) in central Maldives. To account for any variation across day times, bioerosion rates for each size class of each species were then estimated for morning (sunrise–11:30), midday (11:30–14:00), and afternoon (14:00–sunset) using time period averaged bite rates (following the calculations described in Yarlett et al. 2018 [44]). Bioerosion rates for each size class of each species observed during each video were then estimated as:

$$VBSS \; (kg \; survey \; area - 1 \; video \; duration - 1) \; = \; No. \; individuals \; observed \times$$
$$mean \; residence \; time \; (s) \; \times \; bioerosion \; rate \; at \; video \; time \; period \; (kg \; ind - 1 \; s - 1)$$

where VBSS = video bioerosion for each size class of each species.

These values were then converted to bioerosion rates per m$^2$ (per unit time) using the estimated survey area of the video, and then to annual rates per m$^2$ by scaling to the length of the time period (~264 min for morning and afternoon time periods, and ~132 min for midday) and multiplying by 365. This was repeated for all 15 videos in each habitat, covering the whole of the parrotfish feeding day (~11 h; [44]). These time period specific rates were then summed to determine an average annual bioerosion rate (ABR) for each size class of each species.

**Table 1.** For species where data were absent, data for these were assumed to match the bioerosion rate and grazing pressure of the most comparable species for which data were available.

| Species with Missing Data | Data Assumption |
| --- | --- |
| *Chlorurus enneacanthus* | *Chlorurus sordidus* |
| *Scarus tricolor* | *Scarus niger* |
| *Scarus scaber* | *Scarus frenatus* |
| *Scarus prasiognathos* | *Scarus frenatus* |
| *Scarus viridifucatus* | *Scarus frenatus* |
| *Scarus russelii* | *Scarus frenatus* |
| *Hipposcarus harid* | *Scarus frenatus* |
| *Cetoscarus ocellatus* | *Chlorurus strongylocephalus* |
| Juveniles | Lowest measured bioerosion rate at <15 cm |

Finally, total bioerosion rates for each habitat were estimated using the following equation:

$$TAHB \; (kg \; yr - 1) = \sum ABR(kg \; m - 2 \; yr - 1) \times habitat \; area \; (m2)$$

where TAHB = total annual habitat bioerosion. Each variable involved in the calculation had an associated standard error. To calculate cumulative error, standard rules for error propagation were used, with details provided in the Electronic Supplementary Information.

*2.4. Inter-Habitat Variability in Total Parrotfish Grazing Pressure*

To estimate parrotfish grazing pressure, estimates of grazing scar surface areas were derived from the grazing scar length and width measurements used to estimate scar volumes in Yarlett et al. 2018 [44] (surface areas presented in Table 2). The surface area of substrate grazed per minute by different species and size classes was calculated as follows (it was assumed that all bites remove algae from the reef substrate):

$$SAsubstrate \; (cm2 \; min - 1) = \; average \; bite \; rate \; at \; time \; period \; (bpm) \times GSSA \; (cm2)$$

where SAsubstrate = surface area of substrate grazed per minute, bpm = bites per minute for the specific species size class, and GSSA = grazing scar surface area for the specific species size class.

**Table 2.** Mean grazing scar surface areas (cm$^2$) and standard errors (SE) for four size classes of five representative Maldivian parrotfish species. Note that some individuals of *Scarus frenatus* were observed up to ~50 cm, but grazing scar surface areas were assumed to match those in the 31 to 45 cm size class.

| Species | Size Class | N | Mean | SE |
|---|---|---|---|---|
| *Chlorurus sordidus* | <15 cm | 13 | 0.03 | 0.01 |
| | 16 to 30 cm | 22 | 0.05 | 0.01 |
| | 31 to 45 cm | 7 | 0.19 | 0.05 |
| | >46 cm | N/A | N/A | N/A |
| *Chlorurus strongylocephalus* | <15 cm | 13 | 0.03 | 0.01 |
| | 16 to 30 cm | 19 | 0.17 | 0.03 |
| | 31 to 45 cm | 12 | 0.57 | 0.11 |
| | >46 cm | 12 | 0.88 | 0.17 |
| *Scarus frenatus* | <15 cm | 6 | 0.02 | 0.01 |
| | 16 to 30 cm | 11 | 0.04 | 0.01 |
| | 31 to 45 cm | 10 | 0.10 | 0.02 |
| | >46 cm | N/A | N/A | N/A |
| *Scarus niger* | <15 cm | 14 | 0.01 | 0.003 |
| | 16 to 30 cm | 12 | 0.05 | 0.01 |
| | 31 to 45 cm | 9 | 0.08 | 0.02 |
| | >46 cm | N/A | N/A | N/A |
| *Scarus rubroviolaceus* | <15 cm | 4 | 0.01 | 0.003 |
| | 16 to 30 cm | 7 | 0.02 | 0.003 |
| | 31 to 45 cm | 12 | 0.08 | 0.03 |
| | >46 cm | 7 | 0.15 | 0.04 |

The surface area grazed for each size class of each species observed during the video was then estimated using the following equation:

$$VGSS\ (cm2\ survey\ area - 1\ video\ duration - 1)$$
$$= No.\ individuals\ observed\ \times mean\ residence\ time\ (s)$$
$$\times SA\ substrate\ at\ video\ time\ period\ (cm2\ s - 1)$$

where VGSS = video grazed area for each size class of each species.

These values were then converted to area grazed per m$^2$ of substrate using the estimated survey area of the video, and then scaled to the length of the time period (morning, midday, or afternoon, which together make up ~11 h, the length of the feeding day; [44]). This was multiplied by 365 to give an annual rate and was repeated for all 15 replicate videos in each habitat before summing the average morning, midday, and afternoon rates to find an average Annual Grazing Pressure (AGP) for each size class of each species. The total surface area grazed by parrotfish in each habitat was then estimated using the following equation:

$$TAGA\ (cm2\ yr - 1)\ =\ \sum AGP\ (cm2\ m - 2\ yr - 1)\ \times habitat\ area\ (m2)$$

where TAGA = total area grazed annually.

The parrotfish grazing pressure in each habitat was expressed as a proportion of TAGA to the area of substrate available for feeding in each habitat. The surface area available for feeding was estimated using the total surface area and percent cover of dead coral and rubble substrates in each habitat (where habitat surface area was calculated from the 2D spatial extent, extracted from the habitat map (Figure 1), and multiplied by its average rugosity—extracted from Perry et al. 2017 [28]).

## 3. Results

Fifteen species of parrotfish from five genera were identified over six of the eight delineated habitats on the Vavvaru reef platform. No parrotfish were observed in the central nearshore sand/rubble or lagoonal sand habitats. Of these fifteen species, four were excavators (*Chlorurus sordidus, C. strongylocephalus, C. enneacanthus* and *Cetoscarus ocellatus*), and ten were scrapers (*Scarus psittacus, S. frenatus, S. rubroviolaceus, S. niger, S. tricolor, S. russellii, S. prasiognathos, S. scaber, S. viridifucatus,* and *Hipposcarus harid*). One browser species (*Calatomus carolinus*) was also observed and recorded but was not factored into substrate bioerosion or grazing calculations because it feeds on macroalgae rather scraping or excavating the reef substrate.

### 3.1. Species Contributions to Bioerosion and Inter-Habitat Variability in Bioerosion Rates

As predicted, bioerosion was dominated in all habitats (except the nearshore lagoon—Z3) by excavating species (Figure 2; see Tables S9–S14 for these data expressed as rates and standard errors). In the western hardground (Z4) and rubble (Z5) habitats, *C. strongylocephalus* was responsible for >80% of the total parrotfish bioerosion rate (0.42 ± 0.12 and 0.72 ± 0.11 kg m$^{-2}$ yr$^{-1}$, respectively) and was also dominant (albeit to a lesser extent) in the southeast (SE) patch reef (Z1) habitat (>60%; 0.35 ± 0.07 kg m$^{-2}$ yr$^{-1}$). These rates were almost entirely the result of bioerosion by large (>30 cm) individuals. The *Porites* bommie habitat (Z6), on the western side of the platform, was an exception because no *C. strongylocephalus* were observed. Instead, *C. enneacanthus* was responsible for >50% of the total parrotfish bioerosion rate (0.04 ± 0.01 kg m$^{-2}$ yr$^{-1}$). In the northeast (NE) reefs (Z2), *Ce. ocellatus* and *C. sordidus* were the dominant bioeroders (0.22 ± 0.06 and 0.15 ± 0.02 kg m$^{-2}$ yr$^{-1}$, respectively). The nearshore lagoon (Z3) was the only habitat where scrapers eroded more framework than excavators but overall erosion rate in this habitat was low (0.005 ± 0.0009 kg m$^{-2}$ yr$^{-1}$).

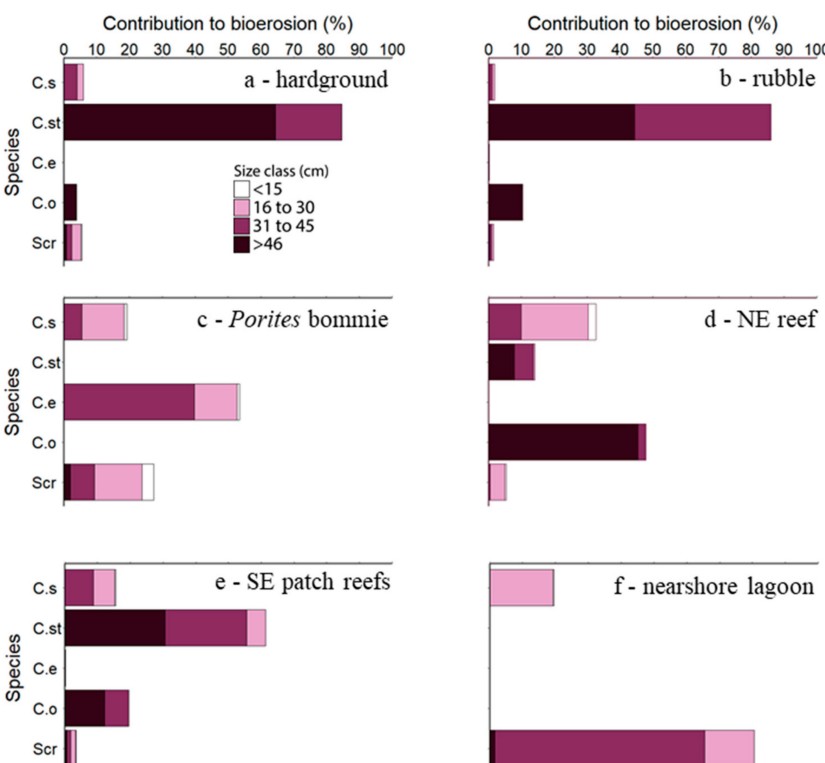

**Figure 2.** Percent contributions to total parrotfish bioerosion by four size classes of the fifteen species present in the six Vavvaru habitats supporting parrotfish. Species abbreviations: C. s—*Chlorurus sordidus*, C. st—*C. strongylocephalus*, C. e—*C. enneacanthus*, C. o—*Cetoscarus ocellatus*, Scr—Scrapers (pooled).

Parrotfish bioerosion rates differed markedly among habitats over the Vavvaru reef platform, ranging from 0.00 to 0.84 ± 0.12 kg m$^{-2}$ yr$^{-1}$ (Figure 3, Table 3). Over half of total platform-scale parrotfish bioerosion occurred in the rubble habitat (Z5), despite this habitat making up only ~12% of the platform area. The NE reef (Z2) and SE patches (Z1) also had high total parrotfish bioerosion rates at 0.46 ± 0.07 and 0.58 ± 0.07 kg m$^{-2}$ yr$^{-1}$, respectively. Approximately 20% of total parrotfish bioerosion over the platform occurred in these habitats combined. Parrotfish were not found in the central nearshore sand/rubble (Z7) and lagoonal sands (Z8) habitats, so they were considered unlikely to make any meaningful contribution to substrate bioerosion in over half of the platform area.

**Table 3.** Total parrotfish bioerosion rate (mean ± SE) and the % that occurs in the morning (sunrise–1130), at midday (1130–1400), and in the afternoon (1400–sunset). Note that midday is a shorter time period (by half) of the morning and afternoon time periods. Total habitat bioerosion (mean ± SE) and the relative % of total platform bioerosion that occurs in each of the eight habitats is also presented. The relative habitat sizes (in % of platform area) are shown for reference. Z1-8 refers to the reef zones marked out in Figure 1.

| | Z1 | Z2 | Z3 | Z4 | Z5 | Z6 | Z7 | Z8 |
|---|---|---|---|---|---|---|---|---|
| Total parrotfish bioerosion rate (kg m$^{-2}$ yr$^{-1}$) | 0.58 ± 0.07 | 0.46 ± 0.07 | 0.01 ± 0.00 | 0.50 ± 0.11 | 0.84 ± 0.12 | 0.08 ± 0.01 | 0.00 ± 0.00 | 0.00 ± 0.00 |
| Morning bioerosion (% of total rate) | 41 | 37 | 62 | 31 | 77 | 53 | N/A | N/A |
| Midday bioerosion (% of total rate) | 6 | 16 | 12 | 4 | 21 | 4 | N/A | N/A |
| Afternoon bioerosion (% of total rate) | 53 | 47 | 27 | 65 | 2 | 43 | N/A | N/A |
| Total parrotfish habitat bioerosion (kg yr$^{-1}$) | 8413 ± 1069 | 23,872 ± 3400 | 280 ± 50 | 34,143 ± 7634 | 80,503 ± 11122 | 6351 ± 789 | 0 ± 0 | 0 ± 0 |
| % of total platform parrotfish bioerosion | 5.48 | 15.55 | 0.18 | 22.23 | 52.42 | 4.14 | 0.00 | 0.00 |
| % platform area | 1.74 | 6.19 | 6.53 | 8.25 | 11.56 | 9.68 | 28.93 | 22.12 |

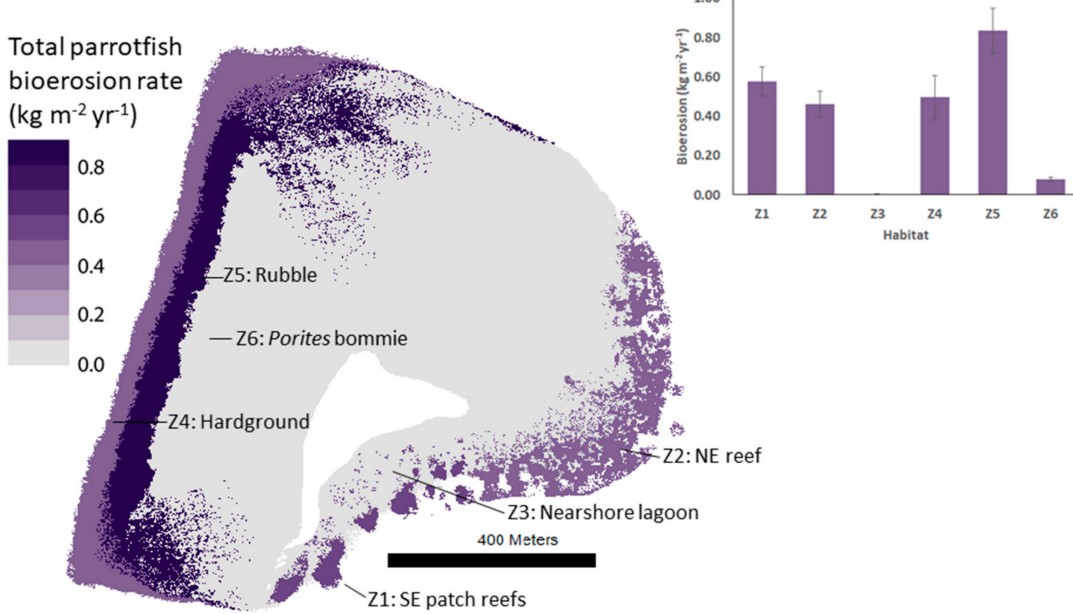

**Figure 3.** Choropleth map showing total rate of parrotfish bioerosion in each marine habitat on the Vavvaru platform. Bar graph subplot shows habitat bioerosion rates and standard error.

### 3.2. Species Contributions to Grazing and Interhabitat Variability in Grazing Pressure

In comparison to bioerosion, a wider variety of parrotfish species and size classes made significant contributions to grazing (Figure 4; see Tables S15–S20 for data expressed as surface area grazed and standard errors). Both scrapers and excavators contributed to substrate grazing, but scrapers grazed a higher surface area compared to excavators in three of the six habitats occupied by parrotfish (Hardground (Z4), *Porites* bommie (Z6), and nearshore lagoon (Z3); Table 4). Some highly abundant species, such as *S. psittacus,* which contributed extremely little to bioerosion, were very important for grazing large surface areas of reef substrate in some habitats (e.g., in the Hardground (Z4) and *Porites* bommie (Z6) habitats).

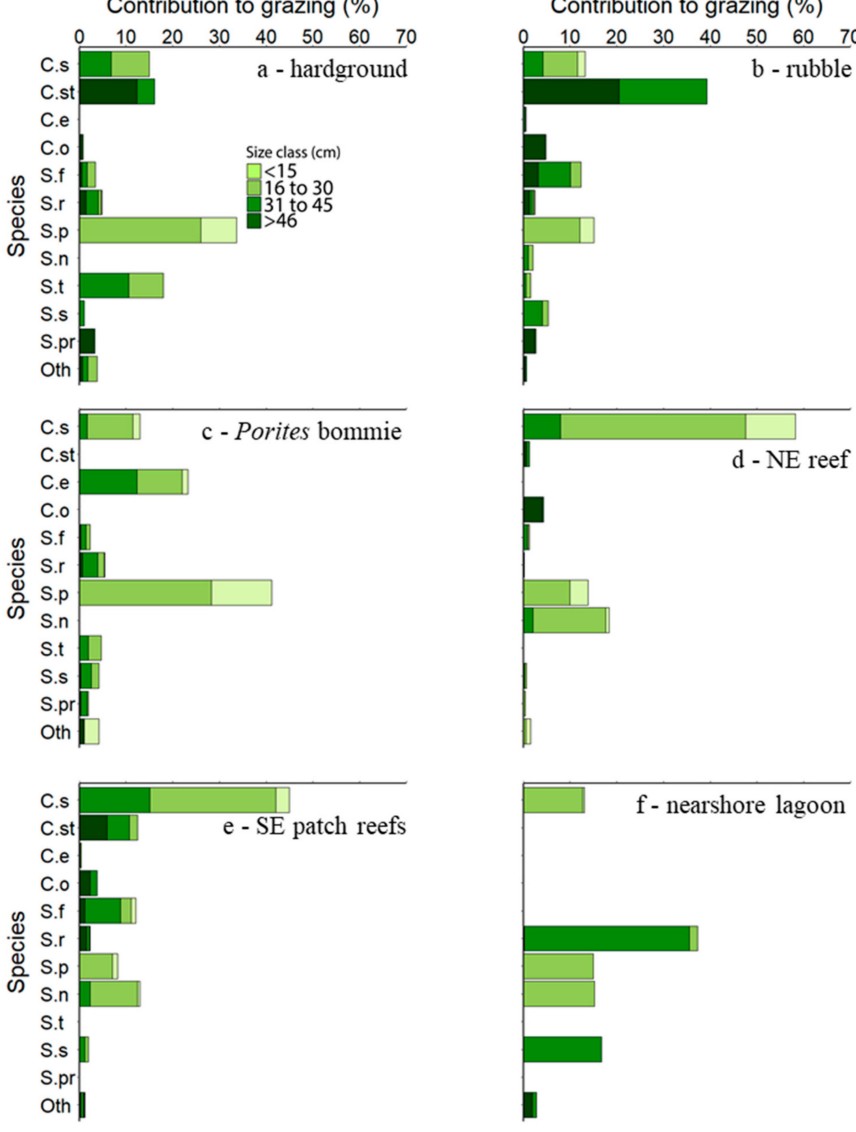

**Figure 4.** Percent contributions to total grazing by each size class of fifteen parrotfish species present in the six major Vavvaru reef habitats. Species abbreviations: C. s—*Chlorurus sordidus*, C. st—*C. strongylocephalus*, C. e—*C. enneacanthus*, C. o—*Cetoscarus ocellatus*, S.f—*Scarus frenatus*, S. r—*S. rubroviolaceus*, S. p—*S. psittacus*, S. n—*S. niger,* S. t—*S. tricolor*, S. s—*S. scaber*, S. pr—*S. prasiognathos*, Oth.—Other.

**Table 4.** Surface area of each habitat and the % of substrate available for feeding. The total area of substrate grazed by parrotfish per year in each habitat is presented, along with the % that occurs in the morning (sunrise–11:30), at midday (11:30–14:00), and in the afternoon (14:00–sunset). Note that midday is a shorter time period (by half) of the morning and afternoon time periods. The percentage of the substrate available for feeding that is grazed per year is also presented. Z1-8 refers to reef zones marked out in Figure 1.

| | Z1 | Z2 | Z3 | Z4 | Z5 | Z6 | Z7 | Z8 |
|---|---|---|---|---|---|---|---|---|
| **Habitat Surface Area (m$^2$) (Area × Rugosity)** | 32,449 | 102,233 | 59,367 | 92,150 | 159,990 | 107,405 | 255,725 | 191,750 |
| **Substrate available for feeding %** | 42.84 | 40.87 | 21.31 | 61.89 | 80.02 | 62.38 | N/A | N/A |
| **Total area grazed by parrotfish (m$^2$ yr$^{-1}$)** | 18,112 ± 1190 | 110,218 ± 6864 | 3826 ± 465 | 74,054 ± 5283 | 72,722 ± 5531 | 75,696 ± 4689 | 0 ± 0 | 0 ± 0 |
| **Morning grazing (% of total rate)** | 25 | 38 | 67 | 43 | 57 | 52 | N/A | N/A |
| **Midday grazing (% of total rate)** | 18 | 17 | 11 | 15 | 12 | 8 | N/A | N/A |
| **Afternoon grazing (% of total rate)** | 57 | 45 | 22 | 42 | 31 | 40 | N/A | N/A |
| **% of habitat area grazed by parrotfish per year** | 130 ± 9 | 264 ± 16 | 30 ± 4 | 130 ± 9 | 57 ± 4 | 113 ± 7 | 0 | 0 |

The surface area of substrate grazed by parrotfish also differed among reef habitats but followed a different pattern to that of bioerosion (Figure 5). Parrotfish grazing pressure was highest in the NE reef (Z2) habitat (~264 ± 16% available substrate grazed yr$^{-1}$). Relative to substrate available for feeding, grazing pressure on reef habitats was comparable in the *Porites* bommie (Z6), SE reef (Z1), and Hardground (Z4) habitats (*Porites* bommie: 113 ± 7%, SE reef: 130 ± 9%, Hardground: 130 ± 9% available substrate grazed yr$^{-1}$) but lower in the Rubble (Z5) habitat (Rubble: 57 ± 4% available substrate grazed yr$^{-1}$; Table 4).

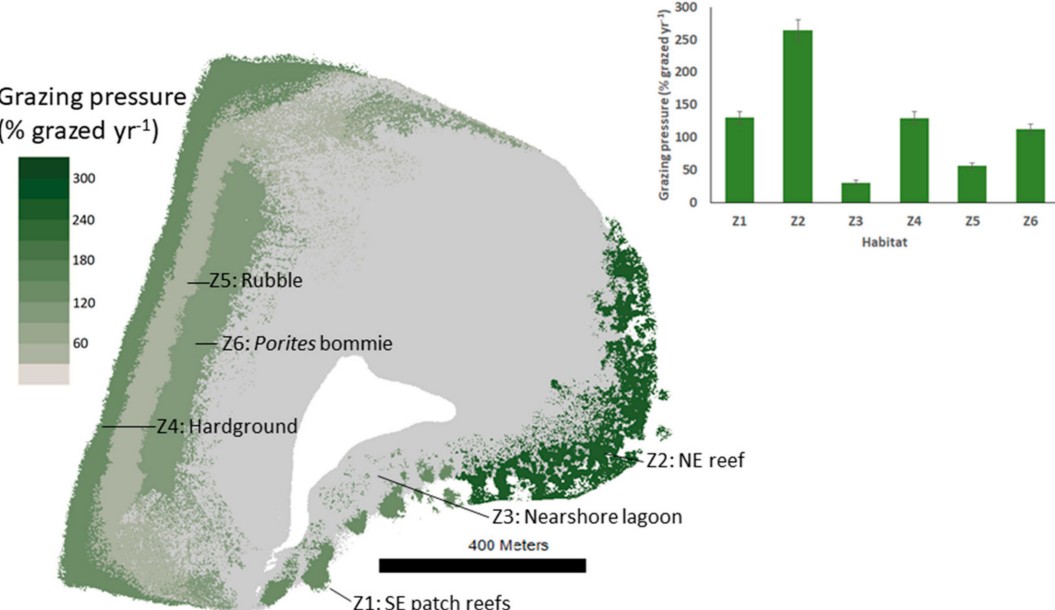

**Figure 5.** Choropleth map showing total parrotfish grazing pressure on the Vavvaru platform. Bar graph subplot shows grazing intensity and standard error.

## 4. Discussion

Our findings demonstrate the extent to which parrotfish bioerosion rates and grazing pressure can vary across different reef habitats and that the spatial patterns of these ecological functions are not necessarily tightly coupled. This means that a habitat with a high parrotfish grazing pressure does not necessarily also have a high parrotfish bioerosion rate, or vice versa. Instead, grazing pressure and bioerosion rate are determined by the species and sizes of parrotfish present in a habitat and what ecological functions they contribute to. These different spatial patterns are important to consider because both ecological functions have different contributions to the reef system. Whereas mapping out spatial patterns of parrotfish bioerosion rates can help identify key sources of platform sediment production and can be used in estimates of reef carbonate budgets, patterns of grazing pressure may indicate the type of habitats where parrotfish act as important controls on benthic algal communities.

### 4.1. Spatial Patterns of Parrotfish Bioerosion Rate and Grazing Pressure

Over 90% of total platform-scale parrotfish bioerosion occurred in the hardground (Z4), rubble (Z5), and NE reef (Z2) habitats, despite comprising only 26% of the total platform area when combined. Over 50% of this bioerosion occurred in the rubble habitat (Z5) alone. This habitat was considered to be naturally important for parrotfish bioerosion and resultant sediment production, rather than representing a "disturbed" rubble-dominated reef front habitat, which has been observed to be exploited by excavating parrotfish [46,47]. Whereas previous research has suggested that parrotfish are not ecologically important in rubble habitats (because the conditions are poor for corals, e.g., see Adam et al. 2015 [48]), our results suggest that these habitats may supply significant quantities of sediment to reef habitats and islands because of high parrotfish bioerosion rates ($0.84 \pm 0.12$ kg m$^{-2}$ yr$^{-1}$). The important contribution of the rubble habitat (Z5) at Vavvaru was partly attributed to the high overall parrotfish bioerosion rate, which was primarily a result of *C. strongylocephlaus* feeding (>80% of total parrotfish bioerosion), but also due to the fact that it was the largest reef habitat in which parrotfish were found (96380 m$^{-2}$; ~12% of the platform area). The NE reef and SE patch reef habitats also had reasonably high overall bioerosion rates ($0.46 \pm 0.07$ and $0.58 \pm 0.07$ kg m$^{-2}$ yr$^{-1}$, respectively), but because of their relatively small spatial extent (51,633 and 14,551 m$^{-2}$, respectively), the total quantity of framework eroded per year in these habitats was lower than that of the rubble habitat. There are, of course, other bioeroding organisms that contribute to total bioerosion rates (such as other fish groups, sponges, urchins, and boring mollusks), but these appear to contribute little to total bioerosion at most sites in the Maldives [26,28].

The observed spatial variation in parrotfish grazing pressure was driven primarily by the contributions of scrapers and small excavators as well as the proportion of substrate available for feeding (rubble and dead coral skeletons). For example, parrotfish grazing pressure was distributed over a large area in the rubble (Z5) habitat ($72,722 \pm 5531$ m$^2$ grazed yr$^{-1}$ distributed over 128,024 m$^2$) compared to the NE reef (Z2), which received the highest parrotfish grazing pressure on the platform ($110,218 \pm 6864$ m$^2$ grazed yr$^{-1}$ distributed over 41,783 m$^2$). The parrotfish grazing pressure in the NE reef (Z2) habitat (which we estimate is fully grazed 2.6-times per year) was comparable to that reported for similar reef crest environments surrounding rat-infested islands (with low seabird density and hence limited nutrient input into surrounding waters) in the neighboring Chagos Archipelago (which is grazed up to 2.8-times per year by parrotfish [49]). The grazing pressure measured in the present study was considerably lower than that reported around seabird-dominated (nutrient-enriched) islands in the Chagos Archipelago, and on inner- and mid-shelf reefs on the northern Great Barrier Reef, which were fully grazed in the region of 9- to 11-times per year [25–49]. The lower rates in our study may be because of low external nutrient input to the reef, such as from seabirds [49] or from terrestrial environments (such as may be the case on inner-shelf reefs on the Great Barrier Reef), and the resultant impacts on substrate food resources. The lower grazing pressures (compared to the NE reef—Z2) in other Vavvaru habitats may be because of the greater surface area of substrate available for feeding (60–80% of habitat surface area in the western habitats). In these habitats, parrotfish

grazing is spread over a large grazeable area compared to the NE reef (Z2), where higher coral cover means parrotfish grazing is condensed into only ~41% of the habitat area. However, parrotfish are not the only abundant grazers on the Vavvaru platform. It is also likely that other families, such as surgeonfish and rabbitfish, have a significant influence on grazing pressure, and potentially their own unique inter-habitat patterns, although this was not investigated. Finally, lower grazing rates do not necessarily translate into altered benthic dynamics, since less productive reefs require lower grazing pressure to keep macroalgae under control.

### 4.2. Factors Influencing the Observed Spatial Patterns

The key contributors to grazing pressure were observed to be different to the key contributors to bioerosion rates and were spread across a larger number of species and size classes. For example, key species in the NE reef (Z2) were found to be the small excavator *C. sordidus* and scrapers *Scarus niger* and *S. psittacus*, which contributed 59%, 18%, and 14%, respectively. Scrapers were also found to make a higher contribution to grazing pressure than excavators in three habitats (the hardground (Z4), *Porites* bommie (Z6), and nearshore lagoon (Z3) habitats). Whereas the bites of scrapers may be smaller, the bite rate for many scraping species in the Maldives, as well as other locations, is considerably higher [15–51], contributing to the high surface area grazed. However, the fact that contributions to grazing are spread across a larger number of species compared to bioerosion does not necessarily infer functional overlap. For example, Brandl and Bellwood (2014) [51] found that different species, even those closely related such as *S. frenatus* and *S. oviceps* [45], utilize different microhabitats for feeding.

The patterns of parrotfish bioerosion rate and grazing pressure observed in the present study are a result of the species assemblages in each habitat. These assemblages are known to be influenced by habitat characteristics such as benthic community composition, nursery habitat availability, substrate type, and degree of structural complexity [52–54]. Factors such as depth, exposure, and distance from the reef slope can also play a role [47–56], as they are known to influence wave energy regimes and currents, which have a resultant impact on fish swimming performance and assemblages [57–59]. The distance from the reef slope and shallow depth may explain why the excavator *C. enneacanthus* was found almost exclusively in the *Porites* bommie (Z6) habitat at Vavvaru. This species may thrive here, while the dominant excavator in the platform edge habitats was found to be *C. strongylocephalus*. Biotic factors, including competition and predation, are also likely to affect assemblage composition [60,61].

Peak parrotfish bioerosion rates and grazing pressure were also observed to occur at different times of the day in different habitats. As a general trend, grazing pressure appeared to be higher in the afternoons at the two main eastern reef habitats but was higher in the mornings in the western hardground (Z4), rubble (Z5) and *Porites* bommie (Z6) habitats (see Tables 3 and 4). This trend was similar for bioerosion, except for in the hardground (Z4) habitat, where higher bioerosion rates occurred in the afternoon rather than the morning (it is also worth noting that daily averages of bioerosion and grazing pressure gave almost the same annual rates as calculations factoring for time period, and suggest that daily variation could be ignored if necessary for estimating annual rates in future studies). We hypothesize that this pattern may reflect the nutritional quality of autotrophic food resources at different times of the day on the eastern and western sides of the platform, but without relevant data, this remains speculative. To further understand the drivers of parrotfish distributions and of these ecological functions that result from parrotfish feeding, there is a need for more research into parrotfish resource harvesting and the partitioning of food resources across species over both spatial and temporal scales. Emerging research in this field has shown that different parrotfish species feed on substrates at different stages of taphonomic succession, even though all target microscopic photoautotrophs, particularly cyanobacteria, as a primary food source [18]. There is also evidence of within habitat spatial variability in bite frequency [40–62], but habitat specific bite rates were not observed in our study. There are two additional areas that warrant further study to refine estimates of bioerosion rate and grazing pressure. First, it was assumed in the present study that all bites remove algae, but future research would benefit from examining the variation in algal biomass removed per

bite. Second, the bioerosion rate and grazing pressure of some species were absent, and were assumed to match that of other species for which data were available. Future research would benefit from directly measuring bite rates and grazing scar metrics of a more diverse range of species to improve the accuracy of reef-scale bioerosion rate and grazing pressure estimates.

*4.3. Implications*

Findings from the present study may help to predict the responses of key parrotfish ecological functions to projected environmental change scenarios, such as habitat degradation. For example, a loss of structural complexity is likely to have a detrimental effect on parrotfish density [63–65]. The results of our study suggest that such impacts would particularly negatively affect key grazing species such as *S. niger* and *S. viridifucatus* that appeared to be associated with topographically complex habitat types at Vavvaru, while other species may remain relatively unimpacted. This change may result in an increase in algal biomass and reduced coral recruitment, but increased bioerosion rate. This pattern has been observed at sites in the southern Maldives, which suffered up to 75% coral mortality during the 2016 bleaching event in habitats comparable to the eastern reef habitats at Vavvaru [66]. The result of this event was an increase in parrotfish bioerosion and pulses of increased sediment generation after subsequent bleaching events, thought to be the result of increased availability of food resources following coral mortality [66–69].

Our findings may also be useful for considering the impact of fishing pressure on the functions of bioerosion and grazing in reef habitats. Parrotfish are unsustainably exploited in many island settings [70]. Large excavators (such as large *Chlorurus* spp.) are typically extracted [71–73] and have been observed to decline in abundance along a gradient of human fishing pressure in some locations, resulting in marked declines in bioerosion rates (e.g., Bellwood et al. 2012 [72]). Although speculative and in need of empirical study, this removal of large excavators could also reduce rates of sediment production in habitats where excavators are the dominant bioeroders, which may have a negative impact on reef island maintenance, especially under projected rates of sea level rise [26]. Efforts should therefore be made to conserve parrotfish and their ecological functions, both through protecting a diverse range of habitat types and through creating refuges from fishing pressure.

**Data Access**

The research data supporting this publication are openly available from the University of Exeter's institutional repository at: https://doi.org/10.24378/exe.2683.

**Supplementary Materials:** The following are available online at http://www.mdpi.com/1424-2818/12/10/381/s1, Figure S1: Example images from the eight distinct Vavvaru marine habitats. Green boxes represent survey area. Table S1: Summary of environmental variables defining the six delineated habitats that supported parrotfish populations on the Vavvaru platform. Table S2: Rate of parrotfish occurrence in the Hardground habitat. Table S3: Rate of parrotfish occurrence in Rubble habitat. Table S4: Rate of parrotfish occurrence in *Porites* bommie habitat. Table S5: Rate of parrotfish occurrence in Nearshore Lagoon habitat. Table S6: Rate of parrotfish occurrence in NE reef habitat. Table S7: Rate parrotfish occurrence in SE reef habitat. Table S8: Number of parrotfish species and total parrotfish occurrence in videos in each habitat. Table S9: Overall parrotfish bioerosion rates in Hardground habitat. Table S10: Overall parrotfish bioerosion rates in Rubble habitat. Table S11: Overall parrotfish bioerosion rates in *Porites* bommie habitat. Table S12: Overall parrotfish bioerosion rates in NE reef habitat. Table S13: Overall parrotfish bioerosion rates in SE patch reefs habitat. Table S14: Overall parrotfish bioerosion rates in Nearshore Lagoon habitat. Table S15: Overall parrotfish grazing pressure in Hardground habitat. Table S16: Overall parrotfish grazing pressure in Rubble habitat. Table S17: Overall parrotfish grazing pressure in *Porites* bommie habitat. Table S18: Overall parrotfish grazing pressure in NE reef habitat. Table S19: Overall parrotfish grazing pressure in SE patch reefs habitat. Table S20: Overall parrotfish grazing pressure in Nearshore Lagoon. Table S21: Average "morning" bites per minute for representative parrotfish species. Table S22: Average "midday" bites per minute for representative parrotfish species. Table S23: Average "afternoon" bites per minute for representative parrotfish species.

**Author Contributions:** All authors were involved in the design of the research. R.T.Y., C.T.P. and R.W.W. collected field data. R.T.Y. and A.R.H. analysed the data. R.T.Y. wrote the draft of the manuscript and all authors contributed to final version. All authors have read and agreed to the published version of the manuscript.

**Funding:** The study was funded by a GW4+ Doctoral Training Partnership studentship from the Natural Environment Research Council to RTY (NE/L002434/1).

**Acknowledgments:** Fieldwork was conducted under a Ministry of Fisheries and Agriculture permit (number 30-D/INDIV/2014/2363). The suggestions of Nick Graham, Jamie Stevens and three anonymous reviewers helped strengthen the manuscript. Neville England kindly constructed the camera frames used in the study. We thank the staff at Korallionlab Marine Research Station for their support during fieldwork. The assistance of Kate Philpot in the field was also greatly appreciated. This is contribution #219 from the Coastlines and Oceans Division in the Institute of Environment at Florida International University.

**Conflicts of Interest:** The authors declare no conflict of interest.

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
