# Peer review of "Inter-Habitat Variability in Parrotfish Bioerosion Rates and Grazing Pressure on an Indian Ocean Reef Platform"

_diversity, doi:10.3390/d12100381_

Round 1
Reviewer 1 Report
This is a very complete submisson of the work carried out for this study. The authors provide clear explanations on the background, procedures and results.
It would be useful to the reader if the Discussion could be broken up via sub-headings.
The imposition of the potential impacts of fishing pressure were not part of the study and are therefore purely speculative. The authors should consider removing this section as fishing (effort, power) for parrotfish varies considerably throughout the Indo-Pacific Region, as do the dominant parrotfish species in each particular jurisdiction. Evaluating the importance of fishing pressure on bioerosin would really require local studies to avoid generalizations that may unnecessarily result in the loss of artisanal fishing opportunities.
While it is understood that several references are germane to cite there are several instances of over citation in the manuscript. Most citations really should be limited to 3 references, and in lines 54, 56, 62, 66, 341, 350, 355 the authors have exceed that limit.
Figure 1. The reader cannot distinguish color differences between Z1 and Z2 as they appear to be the same. This needs to be resolved prior to final submittal as these habitats prove important in the results.
Table 1. While the intent of this assumption is well taken, it imparts uncertainty in the results which is not considered again in the manuscript.
Reviewer 2 Report
It is well done field study. Obtained results are interesting and valuable. Paper is logic and well written.
I have just few remarks and suggestions.
1. Why only some parrotfish grazers are substrate bioeroders but not all? (lines 42-43) It should be explained.
2. Is it misprint in the phrase "annual bioerosion erosion rate"? (line 165)
What about fluctuation of parrotfish feeding rate through the year in Maldives? 3. It is pointed in the Materials and Methods that "Field data were collected in early 2015" - in what months?
4. Fish feeding styles as excavator, scraper, browser are noted many times through the text. I suggest to give definition of these term. It is needed because many readers may not know exactly what is difference between these groups of fish.
5. More details should be provided concerning description of habitats studied. It is important for readers who never been in Maldive atolls. What is the "hardground", "rubble", etc.
6. The title for Table 1 should be corrected - it is difficult to understand the title without the text.
7. The correlation between grazing and bioerosion should be dicussed more clear.
Reviewer 3 Report
The paper is interesting and well written. Please see below for specific comments.
General. Authors seem to use the words "community" and "assemblage" interchangeably. I suggest reading the following paper to distinguish them.
Stroud, James T., Michael R. Bush, Mark C. Ladd, Robert J. Nowicki, Andrew A. Shantz, and Jennifer Sweatman. 2015. 'Is a community still a community? Reviewing definitions of key terms in community ecology', Ecology and Evolution, 5: 4757-65.
Line 19. Please rephrase "the platform margin coral rubble habitat." It is difficult to understand.
Line 95. Please clarify what you mean by "an atoll edge reef platform." A reef located at the edge of an atoll? Also, what is "reef platform"? Is it not just a reef? What is the difference.
Line 114. Please clarify how many habitats in total you had and what they were in the text (not just Figure 1).
Line 165. Is "annual bioerosion erosion rate" supposed to be "annual bioerosion rate"?
Line 168. "TAHB (kg habitat area-1 yr-1)" This expression is confusing. I assume you mean "kg yr-1" for the total area of the habitat, then the unit should be just "kg yr-1."
Lines 173-175. This sounds like the authors made the actual in-situ observations for the present study, but I assume the data was derived from previously-published work (as indicated by "derived from data in Yarlett et al. 2018). Please clarify.
Line 199. "TAGA (cm2 habitat area-1 yr-1)" Again, I have the same comment as TAHB in Line 168.
Line 211. "One browser species (Calatomus carolinus) was also observed and recorded but was not factored into substrate bioerosion or grazing calculations." Please briefly explain and justify why this species was no included in the analysis.
